# Dysregulation of Metabolism and Proteostasis in Skeletal Muscle of a Presymptomatic Pompe Mouse Model

**DOI:** 10.3390/cells12121602

**Published:** 2023-06-11

**Authors:** Marlena Rohm, Leon Volke, Lara Schlaffke, Robert Rehmann, Nicolina Südkamp, Andreas Roos, Anne Schänzer, Andreas Hentschel, Matthias Vorgerd

**Affiliations:** 1Department of Neurology, BG-University Hospital Bergmannsheil gGmbH, Ruhr-University Bochum, 44789 Bochum, Germany; 2Heimer Institute for Muscle Research, BG-University Hospital Bergmannsheil gGmbH, Ruhr-University Bochum, 44789 Bochum, Germany; 3Department of Neuropediatrics, University Hospital Essen, Duisburg-Essen University, 45147 Essen, Germany; 4Children’s Hospital of Eastern Ontario Research Institute, University of Ottawa, Ottawa, ON K1H 8L1, Canada; 5Institute of Neuropathology, Justus Liebig University, 35390 Giessen, Germany; 6Leibniz-Institut für Analytische Wissenschaften, 44139 Dortmund, Germany; andreas.hentschel@isas.de

**Keywords:** Pompe disease, Gaa^6neo/6neo^ mouse, proteomics, muscular disease

## Abstract

Pompe disease is a rare genetic metabolic disorder caused by mutations in acid-alpha glucoside (GAA) leading to pathological lysosomal glycogen accumulation associated with skeletal muscle weakness, respiratory difficulties and cardiomyopathy, dependent from the GAA residual enzyme activity. This study aimed to investigate early proteomic changes in a mouse model of Pompe disease and identify potential therapeutic pathways using proteomic analysis of skeletal muscles from pre-symptomatic Pompe mice. For this purpose, quadriceps samples of Gaa^6neo/6neo^ mutant (Pompe) and wildtype mice, at the age of six weeks, were studied with three biological replicates for each group. The data were validated with skeletal muscle morphology, immunofluorescence studies and western blot analysis. Proteomic profiling identified 538 significantly upregulated and 16 significantly downregulated proteins in quadriceps muscles derived from Pompe animals compared to wildtype mice. The majority of significantly upregulated proteins were involved in metabolism, translation, folding, degrading and vesicular transport, with some having crucial roles in the etiopathology of other neurological or neuromuscular diseases. This study highlights the importance of the early diagnosis and treatment of Pompe disease and suggests potential add-on therapeutic strategies targeting protein dysregulations.

## 1. Introduction

Pompe disease is a lysosomal hereditary multisystemic metabolic disorder (MIM: 232300). The clinical phenotype includes myopathy which is caused by glycogen accumulation due to bi-allelic pathogenic variants in the *GAA* gene-encoding enzyme alpha-glucosidase (human: GAA, mice: Gaa). The accumulation of glycogen in skeletal muscle cells causes skeletal muscle weakness, but also respiratory difficulties and cardiomyopathy dependent from the GAA residual enzyme activity. Pompe disease shows two different clinical phenotypes: infantile-onset Pompe disease (IOPD) and late-onset Pompe disease (LOPD). IOPD is associated with a more severe and rapid disease progression with onset in infancy caused by a total absence of GAA leading to a cardiorespiratory failure in one year if not treated [1]. LOPD typically presents between the ages of 30 and 60, with residual enzyme activity of 3–30% [2]. Patients with LOPD develop progressive muscle weakness, particularly in skeletal and diaphragmatic muscles, leading to loss of ambulation and often to dependence on assisted ventilation. The regular administration of intravenous enzyme replacement therapy (ERT), which involves recombinant human GAA, can slow the progression of muscle degeneration. However, the optimal timing for initiating ERT in LOPD patients before clinical manifestation remains controversial [3,4].

Mouse models for Pompe disease have been developed and are commonly used to study pathophysiological processes as well as to evaluate new therapeutic approaches. The Gaa^6neo/6neo^ model (Pompe mouse) represents IOPD due to the complete absence of functional Gaa, while the LOPD phenotype is characterized by a late onset of clinical symptoms [5]. The Pompe mice develop progressive motor weakness beginning at the age of 3 to 4 months. Obvious clinical signs of motor impairment, such as waddling gait and muscle weakness, were observed by 7–9 months [6,7].

Clinical proteomics plays an essential role in unravelling the complex biological mechanisms underlying neuromuscular diseases [8]. By analysing the abundance, interaction, function, structure, and post-translational modification of thousands of proteins in a single experiment, proteomics provides unbiased and comprehensive biochemical information that improves our understanding of the disease etiopathology. This information can also help to identify potential targets for developing disease-specific therapies. The term proteostasis describes processes in protein synthesis, folding, trafficking, aggregation and degradation [9,10]. In Pompe disease, disruptions in protein homeostasis, including cellular autophagy, have been identified as major contributors to muscle degeneration in mice [11]. Identifying tissue markers with pathophysiological relevance is a crucial area of research that can shed light on the underlying mechanisms of the disease and may serve as targets for novel or add-on therapies. 

This study aimed to analyse the proteomic signature of the quadriceps muscle of 6-week-old pre-symptomatic Pompe mice and compared them to wildtype controls, to obtain a broad overview of the early dysregulations that contribute to muscle cell vulnerability and subsequent degeneration. In this vein, the study aimed to pinpoint the molecular mechanisms associated with glycogen accumulation and glycogen build-up and, in addition, to go beyond perturbed protein quality control and degradation in terms of early pathobiochemical events triggering muscle cell degeneration. 

## 2. Materials and Methods

### 2.1. Animals

In this study, *Gaa*^6neo/6neo^ mutant mice were used as an animal model for Pompe disease. Mutant animals harbour a targeted disruption in exon 6 of the *Gaa*-gene, which results in the lack of enzyme activity of Gaa [7]. To investigate early histological and associated pathobiochemical changes in the etiology of Pompe disease, four-week-old female Pompe and wildtype mice were obtained (Jackson Laboratory, strain no. 004154, B6;129-*Gaa*^tm1Rabn/J^ with respective wildtypes). Animals were housed in a controlled environment (light/dark cycle 12/12 h; 22 °C; food/water ad libitum) and sacrificed at the age of six weeks. Quadriceps muscles were collected and snap-frozen for histological and proteomic studies. Small samples were fixed in glutaraldehyde for further resin sections and ultra-structural studies.

### 2.2. Sample Processing

A total of 25 slices of 10 µm were cut on a microtome (Leica, Mannheim, Germany) and stored at −80 °C until further processing for analysis by liquid chromatography coupled to tandem mass spectrometry. Here, three biological replicates for wildtype and Pompe mice were used, respectively. The muscle slices of each biological replicate were lysed in 200 µL of 50 mM Tris-HCl (pH 7.8) buffer, 5% SDS, and cOmplete ULTRA protease inhibitor (Roche, Basel, Switzerland) using the Bioruptor^®^ (Diagenode, Denville, NJ, USA) for 10 min (30 s on, 30 s off, 10 cycles) at 4 °C. To ensure complete lysis, an additional sonication step using an ultra-sonic probe (30 s, 1 s/1 s, amplitude 40%), followed by centrifugation at 4 °C and 20,000× *g* for 15 min, was conducted. The protein concentration of the supernatant was determined by a BCA assay according to the manufacturer’s protocol. Disulfide bonds were reduced by the addition of 10 mM TCEP at 37 °C for 30 min, and free sulfhydryl bonds were alkylated with 15 mM IAA at room temperature (RT) in the dark for 30 min. A total of 100 µg protein of each sample was used for proteolysis using the S-Trap protocol (Protifi, Fairport, NY, USA) and using a protein-to-trypsin ratio of 20:1. The incubation time for trypsin was changed to 2 h at 46 °C. Proteolysis was stopped using FA to acidify the sample (pH < 3.0). 

All proteolytic digests were checked for complete digestion after desalting by using monolithic column separation (PepSwift monolithic PS-DVB PL-CAP200-PM, Dionex, Sunnyvale, CA, USA) on an inert Ultimate 3000 HPLC (Dionex, Germering, Germany) by direct injection of 0.5 μg of sample. A binary gradient (solvent A: 0.1% TFA, solvent B: 0.08% TFA, 84% ACN) ranging from 5 to 12% B in 5 min, and then from 12 to 50% B in 15 min at a flow rate of 2.2 Μl/min at 60 °C, was applied. UV traces were acquired at 214 nm [12].

### 2.3. LC-MS/MS Analysis

All samples were analysed by nano LC-MS/MS using 1 µg of sample material. Samples were loaded on an Ultimate 3000 rapid separation liquid chromatography (RSLC) nano system with a ProFlow flow control device coupled to a Qexactive HF mass spectrometer (both from Thermo Fisher Scientific, Waltham, MA, USA). The samples were first separated on a 75 µm × 50 cm, 100, C18 main column with a flow rate of 250 Nl/min and a linear gradient consisting of solution A (99.9% water, 0.1% formic acid) and solution B (84% acetonitrile, 15.9% water, 0.1% formic acid) with a pure gradient length of 120 min (3–45% solution B). The gradient was composed as follows: 3% B for 20 min, 3–35% for 120 min, followed by three washing steps, each lasting 3 min and ranging to 95% buffer B. The instrument was left to equilibrate for 20 min after the final washing procedure. 

### 2.4. Analysis of Proteomic Data

Proteome Discoverer software 2.5.0.400 (Thermo Fisher Scientific, Schwerte, Germany) was used for all data processing, and searches were performed in target/decoy mode against a mouse UniProt database (accessed and downloaded 14 May 2023, UniProt (www.uniprot.org)) using the MASCOT and SEAQUEST algorithms. The following search parameters were used: precursor and fragment ion tolerances of 10 ppm and 0.02 Da for MS and MS/MS, respectively; a trypsin set as enzyme with a maximum of two miss-cleavages; carbamidomethylation of a cysteine set as a fixed modification; and oxidation of methionine set as a dynamic modification and a percolator false discovery rate of 0.01 for PSM, peptide and protein identifications. Label-free quantification (LFQ) analysis was performed for each condition.

The abundance of proteins measurable in at least two wildtype and Pompe mice were taken into account. From those, the significance of abundance was calculated with Students *t*-tests (*p* < 0.05). A volcano plot was generated with R (package ggplot2 v3.4). The fold change in proteins was calculated by dividing wildtype protein abundances with Pompe protein abundances. A Voronoi diagram was generated with the online tool Proteomaps (accessed on 14 March 2023, https://proteomaps.net/) [13]. Here, the tile size correlates with the fold change in each significantly upregulated protein. GO-term analysis and KEGG analysis, in order to identify grouped protein functions and pathways, were carried out with the significantly upregulated proteins using DAVID (accessed on 14 March 2023, https://david.ncifcrf.gov).

### 2.5. Histological Staining

For histochemistry, 5 µm thick frozen sections were processed for haematoxylin–eosin (H&E), Gomori trichrome, nicotinamide adenine dinucleotide (NADH), succinate dehydrogenase (SDH) and cytochrome c oxidase (COX), using standard procedures.

### 2.6. Immunofluorescence Studies

For immunofluorescence-based studies of protein abundances and localization toward the validation of proteomic data, 5 µm cryosections were fixed with acetone for 10 min. Details of antibodies used are shown in Table 1. After blocking with 10% goat serum/1% BSA for one hour at RT, primary antibodies were incubated overnight at 4 °C. After washing with PBS, secondary antibodies were incubated for 1 h at RT. Images were analysed semi-quantitatively, by selecting three random equal-sized fields-of-view from each stained cross-section and thresholding in Fiji, to calculate the positively stained area in µm^2^ [14].

### 2.7. Resin Sections and Electron Microscopy

Quadriceps muscle tissue was processed according to published methods [15,16]. In brief, small samples were fixed in 4% glutaraldehyde/0.4 MPBS and processed according to standard procedures. Sections of 1–2 µm semithin were cut from resin-embedded tissue and stained with Periodic acid-Schiff (PAS) and p-phenylenediamine (PPD). For contrast in transmission electron microscopy (TEM), ultrathin sections were treated with 3% lead citrate-3H_2_O with a Leica EM AC20 (ultrastain kit II) and examined at a Zeiss EM109 TEM equipped with a sharp eye digital camera. 

### 2.8. Western Blot

Frozen mouse quadriceps tissue (~100 mg) was homogenized in ice-cold lysis buffer (NaCl 150 mM, Tris-HCl 10 mM, EDTA 1 mM, Triton X-100 1%, SDS 1%, Sodium deoxycholate 1% and protease inhibitor (Roche)) using ultrasound (Hielscher UP200St ultrasonic with VialTweeter, 20%, interval, 1 min). The muscle lysates were centrifuged for 20 min at 4 °C and 14,000 rcf. The protein concentration of the supernatant was determined using the Micro BCA™ Protein Assay Kit (#23235, Thermo Fisher Scientific, Waltham, MA, USA). The protein solution was mixed with Laemmli buffer (2% SDS, 25% glycerol, 5% 2-mercaptoethanol, 0.01% bromphenol blue, 62.5 mM Tris HCL, pH 6.8) and subsequently denatured for 5 min at 95 °C. Equivalent amounts of protein extracts (20 µg per lane) were separated by SDS-PAGE. One gel was used to measure total protein concentration with Coomassie staining. Another was transferred to a nitrocellulose membrane using a semi-dry system (Trans-Blot^®^ SD Semi-Dry Transfer Cell, Bio-Rad, Hercules, CA, USA). The membrane was blocked using TBS + 1% Tween 20 + 5% milk for 1 h at RT and was then incubated with anti-HSPB7 (ab150390, abcam, Cambridge, UK) at 4 °C overnight. The membrane was washed in TBST and incubated with HRP-conjugated secondary anti-rabbit antibody (G9295, Sigma–Aldrich, St. Louis, MO, USA) for 1 h at RT before washing with TBST. Antibody binding was detected by chemiluminescent substrate Radiance Q (AC2101, Azure Biosystems, Dublin, CA, USA) in the c600 Imaging System (Azure Biosystems, Dublin, CA, USA). The optical densities (OD) of the Coomassie staining were quantified using the Azure Biosystems software (v2.1.097). Fold of protein abundances was normalized to the relative OD in Coomassie staining and data were expressed as relative to wildtype. 

### 2.9. Statistical Analysis

Protein abundances in wildtype and Pompe mice were statistically evaluated by two-sided Student’s *t*-test (* *p* < 0.05, ** *p* < 0.01, *** *p* < 0.001, **** *p* < 0.0001). Statistical testing and graphics were performed with GraphPad Prism (v9.5).

## 3. Results

### 3.1. Histological Staining

The H&E- and Gomori-trichrome-stained skeletal muscle sections from six-week-old Pompe mice exhibited cytoplasmic vacuoles surrounded by darkly stained granules in the majority of the muscle fibres (Figure 1). The normal distribution of the oxidative enzyme activity of the NADH, SDH and COX “checkerboard”, of fast twitch and slow twitch fibres, were not disturbed in Pompe mice. (Figure 1).

### 3.2. Resin Sections and Electron Microscopy

The detailed morphological characteristics of skeletal muscle biopsies were analysed in resin embedded samples from 6-week-old Pompe mice. The PAS-stained resin sections revealed an increase in variation of muscle fibre diameter (Figure 2A). The larger PAS-positive glycogen-containing vacuoles were present in the majority of the muscle fibres, in addition to the smaller PAS-positive vacuoles consistent with lysosomes. In PPD-stained resin sections, the larger vacuoles contained dark lipophilic material indicating autophagosomes (Figure 2B). 

Ultrastructural analyses confirmed the presence of large autophagosomes containing debris and myelin-like bodies (Figure 2C). Glycogen deposits were also detected subsarcolemmal to the myofibres (Figure 2D). Next to the larger autophagosomes, smaller glycogen containing lysosomes were present (Figure 2E).

### 3.3. Overview with GO-Term Analysis

The label-free untargeted proteomic profiling approach enabled the robust quantification of 1251 proteins occurring both in wildtype and Pompe mice, with, remarkably, 538 significantly upregulated and 16 significantly downregulated proteins in quadriceps muscles derived from Pompe animals when compared with wildtype mice (Figure 3B and Appendix A). Notably, Hspb7, a sarcoplasmic cardioprotective chaperone facilitating sarcomeric proteostasis [17], showed the highest increase (fc = 26.08 *p* = 0.0006), whereas Msrb3 (fc = 0.57 *p* = 0.020) displayed the most prominent decrease in the entire dataset. A Voronoi diagram categorized all significantly upregulated proteins, with tile size corresponding to fold change (Figure 3C).

A GO-term-based in silico analysis of dysregulated proteins, focusing on affected biological processes, revealed the involvement of 60 upregulated proteins in translation, 36 in mitochondrial ATP synthesis and lipid metabolic processes, respectively, 31 proteins involved in the negative regulation of apoptotic processes and 17 proteins involved in protein folding. Five of the few downregulated proteins were involved in muscle contraction. Dividing the dysregulated proteins into cellular components with the GO-term analysis, 336 of all dysregulated proteins occurred in the cytoplasm, 224 in the nucleus, 192 in the mitochondria and 70 in the cytoskeleton.

### 3.4. Similarities to Neurodegenerative Diseases

A KEGG analysis revealed similarities to the amyotrophic lateral sclerosis pathway with 91 proteins overlapping, to Alzheimer’s disease and Parkinson’s disease with 87 proteins, as well as to prion disease with 86 and to Huntington’s disease with 86 proteins. For all significantly upregulated proteins contained in the KEGG pathway—“pathway of neurodegeneration—multiple diseases”—see Appendix A. The exemplary central proteins in those diseases are Tdp-43 (Tardbp, fc = 3.87, *p* = 0.04) and Tau (MAPT, fc = 3.33, *p* = 0.005), which were found to be significantly upregulated in the quadriceps of the Pompe mouse.

### 3.5. Altered Proteostasis

The proteomic signature of quadriceps muscle derived from Pompe mice displayed many significantly upregulated proteins involved in translation, folding, sorting and degrading, as well as in vesicular transport (blue and red pathway in the Voronoi diagram, Figure 3C). Those pathways were further summarized under the term “proteostasis”. Among these identified proteins were those which are known to play crucial roles in the etiopathology of other neuromuscular diseases; Fhl1 (fc = 2.88; *p* = 0.01), as a protein shuttling between the sarcoplasm and myonuclei, is involved in muscle development or hypertrophy, and mutations in the corresponding gene were linked to X-linked Emery-Dreifuss muscular dystrophy 6 (EDMD6; MIM: MIM:300696), X-linked dominant scapuloperoneal myopathy (SPM; MIM:300695) and X-linked myopathy with postural muscle atrophy (XMPMA; MIM:300696), respectively. 

Fyco1 (fc = 6.38 *p* = 0.03) is a vesicular protein involved in autophagosome formation and has been linked to vacuole-pathology in sIBM and VCP-related myopathology (MIM:167320), respectively [18,19]. Notably, Vcp was also increased (fc = 1.51; *p* = 0.004). 

Cryab, as part of the small heat shock protein family, with a function to bind to misfolded proteins and prevent protein aggregation, was highly upregulated (fold change in 6.22; *p* < 0.002) [20]. Pathogenic *CRYAB* variants are leading to dominant and recessive forms of myofibrillar myopathy (MFM; 608,810 & 613,869). 

Moreover, Pompe mouse quadriceps showed upregulation of many other proteins involved in ubiquitination and proteolysis such as 20S, 19S, UbCH7/8 and UB. 

### 3.6. Metabolism as a Major Upregulated Factor

Although various histological staining approaches for metabolic activities showed no quantifiable differences between wildtype and Pompe mice, unbiased proteomics revealed many metabolism-associated proteins to be significantly upregulated, as indicated by the orange/brown tiles of the Voronoi diagram (Figure 3C). Among others, subunits of ATP synthase, cytochrome c (validation with immunofluorescence see Figure 4) and NADH dehydrogenase, were found to be upregulated in the quadriceps of Pompe mice, representing upregulated oxidative phosphorylation as indicated by the results of our KEGG analysis. Furthermore, glycogen metabolism pathway proteins Gyg1 (fc = 2.06, *p* < 0.001), Gys2 (fc = 1.38, *p* = 0.003), Gbe1 (fc = 8.63, *p* = 0.001) and Ugp2 (fc = 2.10, *p* < 0.0001) were upregulated in the quadriceps of Pompe mice. This finding not only reflects the general impact of the loss of functional Gaa on glycogen homeostasis in muscle cells, but moreover reflects the robustness of our data set.

To further decipher a molecular interplay of these dysregulated proteins, a STRING analysis was carried out. This in silico approach indicated the central role of Sod1 among the group of unregulated proteins, a finding which, in turn, hints toward the presence of oxidative stress burden in Pompe mouse muscle (Figure 3E).

### 3.7. Immunofluorescence and Western Blot Studies toward Validation of Proteomic Data

To validate our proteomic data, additional immunological-based studies were carried out which were mainly focused on the proteins involved in proteostasis, in addition to CytC and Sod1. These studies confirmed the proteomic findings: CytC and Sod1 both show a diffuse sarcoplasmic increase accompanied by the presence of dot-like structures (Figure 4). The same immunostaining was observed for ubiquitin, Cryab and Fyco1 (Figure 4). The immunofluorescence study of Fhl1 revealed an increase in sarcolemma immunostaining in quadriceps muscle derived from Pompe mice compared to wildtype littermates (Figure 4). Western blot studies of Hspb7, the protein showing the highest abundance in the proteomic signature of Pompe mouse muscle, and an additional validation of Fyco1 and Cryab, also confirmed our mass spectrometry findings with upregulation in Pompe mice (Figure 5).

## 4. Discussion

Pompe disease is a lysosomal glycogen storage disease with a known impact on proper metabolism and proteostasis (control of biogenesis, folding, trafficking and degradation) reflected by the dysregulation of proteins involved in these biological functions [9,10]. Based on proteomic data, we highlight a significant upregulation in metabolism and proteostasis in the quadriceps muscle of six-week-old Pompe mice, even before the onset of clinical symptoms suggesting an early impact of dysregulation of these biological functions in the etiology of Pompe disease.

Although standard histological stainings (NADH, SDH and COX) did not reveal any visible differences suggestive of mitochondrial dysfunction between the Pompe and wildtype quadriceps muscle, the proteomic data clearly indicated metabolic dysregulations reflected by the increased abundances of proteins crucial for mitochondrial function and maintenance. In particular, when comparing the histological COX staining with the results obtained from immunofluorescence studies, the benefits of proteomic analysis became apparent. Proteomics has a heightened sensitivity in contrast to conventional diagnostic methods, especially with regard to identifying early pathophysiological alterations that may become visible at the histological level later in the disease course. 

The clinical phenotype of Pompe disease is caused by the progressive accumulation of glycogen in lysosomes, the enlargement of lysosomes and impaired autophagy, as visualized with PAS-staining, PPD-staining and electron microscopy [16]. The upregulation of proteins involved in glycogen metabolism was detected in Pompe mice and accords with the current literature [21,22]. A proteomic study of skeletal muscle from LOPD patients demonstrated that ERT did not seem to affect the pathways regulating muscle contractile proteins, cytoskeletal rearrangement, lysosome biogenesis and oxidative metabolism. Instead, protein dysregulation persisted after ERT administration [23], in turn suggesting the emergent need for add-on therapies targeting these cellular malfunctions. Indeed, our findings suggest that metabolic pathways, vesicular transport, translation and protein processing are already upregulated at a very early age, before the onset of clinical symptoms in terms of perturbed muscle function. This underlines the importance of early diagnosis and treatment for Pompe disease and highlights the potential value of new-born screening, as already applied in a minority of countries, allowing early therapeutic intervention [24]. As ERT does not correct protein dysregulation as indicated by the previous proteomic study, alternative therapeutic strategies that target protein dysregulations beyond glycogen homeostasis (and subsequent proteolysis), such as enhancing mitochondrial maintenance, may be beneficial.

Remarkably, proteomic data, analysed with KEGG-based in silico analysis, also revealed an intriguing overlap between the pathophysiologies of skeletal muscles of Pompe mice and several other neurodegenerative diseases, including Amyotrophic Lateral Sclerosis (ALS), Alzheimer’s disease, Huntington’s and prions disease. This finding is likely due to the dysregulation of proteins involved in protein processing, such as chaperons and TDP-43, which is a driver of toxic protein aggregation in many neurodegenerative diseases. The accumulation of misfolded proteins in Pompe disease was linked to an autophagic response at the endoplasmic reticulum [25]. TDP-43 accumulations are a hallmark of ALS but aggregates have also been found in the skeletal muscles of patients with limb-girdle muscular dystrophy, inclusion body myositis and oculopharyngeal muscular dystrophy [26,27,28,29]. Thus, our data add Pompe disease to the growing list of myopathies associated with TDP-43 aggregates. The increase in FHL1 in Pompe mouse muscle reflects another link to muscular disorders with reduced body myopathy (based on hemizygous *FHL1* variants) as the most striking example [30,31]. FHL1 appears to serve multiple roles, including acting as a scaffolding protein during sarcomere assembly [32] and interacting with LC3 to participate in the formation of autophagosomes [33]. Unexpectedly, our immunofluorescence data showed FHL1 enrichment at the sarcolemma rather than in central aggregates, suggesting that its role in muscle pathology may be more complex than previously thought. Further investigations are needed to determine the localization of FHL1 in the later stages of the disease when autophagy is enhanced.

Given that pathogenic variants in *GAA* impact lysosomal function as the primary target of Pompe disease, we aimed to gain a comprehensive understanding of this primary etiopathology and therefore also investigated dysregulated proteins with known roles in protein processing, clearance and aggregate formation. Some of these proteins were previously reported to be dysregulated in or directly causative of other muscular diseases. Mutations in *FYCO1*, for example, were associated with inclusion body myositis and were shown to be increased in VCP patients [18,19]. FYCO1 interacts with LC3, suggesting a potential mediator in an autophagic build-up in Pompe disease [34]. CRYAB, a sarcoplasmic chaperone, was previously implicated in myofibrillar myopathies and desmin-related myopathy (DRM), a neuromuscular disease characterized by protein aggregates in muscle cells [20,35]. In drosophila, CRYAB was found to maintain the structural integrity of developing muscles [36]. The paradigmatic key players of modulation of proteostasis, such as FYCO1 and CRYAB, were even confirmed by immunofluorescence studies. Given that these two proteins are involved in protein folding and breakdown, their increase might reflect a cellular attempt to antagonize the build-up of protein aggregates. However, taking the crucial role of CRYAB in the maintenance of structural proteins into consideration, its increase might also be in accordance with an attempt to prevent cytoskeletal breakdown. Further functional studies are needed to prove this assumption. In this vein, it is worth noting that Hspb7 acts as a cardioprotective chaperone facilitating sarcomeric proteostasis [17]. Hspb7 is a member of the small heat-shock protein family and is expressed in cardiac and skeletal muscles. It was shown recently that Hsbp7 is an interacting partner of dimerized filamin C (FLNC) and might be essential for maintaining muscle integrity [37]. Given the high abundance of Hspb7 in the early skeletal muscle of Pompe mice, it might be speculated that Hspb7, interacting with FLNC, is upregulated to prevent sarcomeric insufficiency and the progression of Pompe disease pathology.

The autophagic build-up can also lead to the formation of aggregates (as suggested by the increase in TDP-43) and interfere with other cellular processes and even prevent the therapy from reaching its target [38,39]. The high abundance of FYCO1, a well-known autophagic maker, and the relatively low abundance of upregulated Ubiquitin in the six-week-old Pompe mice, suggest that autophagic degradation is still functioning in the early stages of the disease. This finding may explain the delayed expression of the phenotype, as functional autophagy may contribute to the delay.

## 5. Conclusions

Our combined morphological and biochemical findings suggest that pronounced ultra-structural and molecular changes preceded clinical muscle cell vulnerability. These molecular changes affect protein classes beyond those involved in protein folding and clearance, thus suggesting an already complex pathophysiology before clinical manifestation occurs. Taking this into consideration, one might postulate that, after confirmation and validation of our findings by independent researchers, Pompe disease should be included into the new-born screening and warrants early intervention, ideally before clinical symptoms manifest. 

## Figures and Tables

**Figure 1 cells-12-01602-f001:**
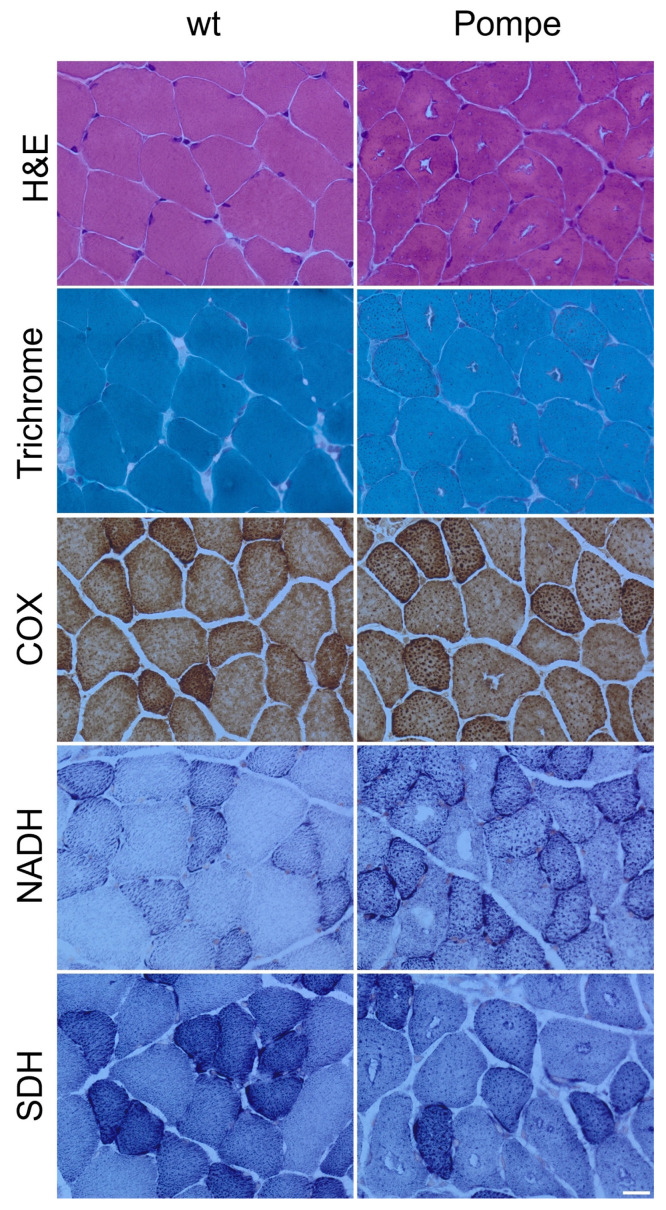
Histopathological studies of the quadriceps muscle cross-sections from 6-week-old Pompe mice revealed vacuoles in H&E and Gomori trichrome staining. Representative staining of COX, NADH and SDH staining showed no difference in 6-week-old Pompe mice compared with age-matched wildtype mice (wt). Scalebar 20 µm.

**Figure 2 cells-12-01602-f002:**
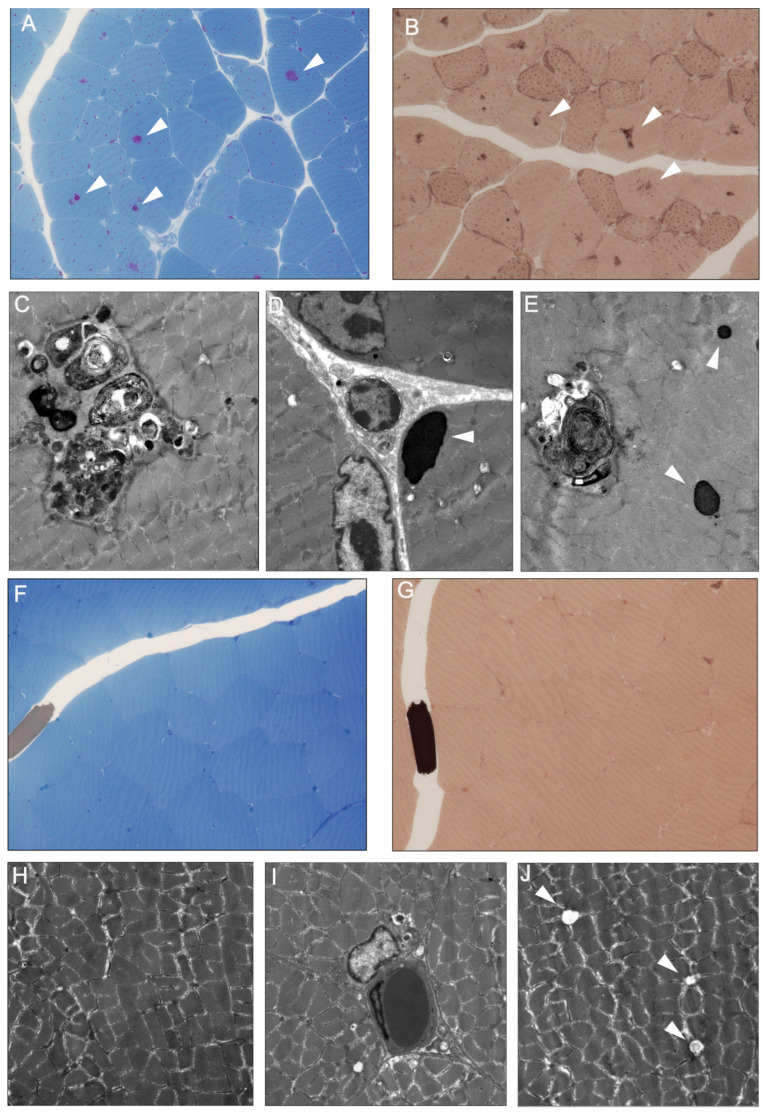
Morphological features of four week old Pompe mice. (**A**) PAS-stained resin sections showed increased diameter of myofibre. PAS-positive larger vacuoles (arrowheads) were present in the majority of myofibre. Small PAS-positive vacuoles were frequent. (**B**) In PPD-stained resin sections, lipophilic material in larger vacuoles were present consistent with autophagosomes (arrowheads). (**C**) Ultrastructural image of large autophagosomes with myelin-like debris inside the vacuoles (**D**) and subsarcolemmal glycogen deposits (arrowhead). (**E**) In addition to large autophagosomes, smaller glycogen containing lysosomes (arrowhead) were present in the myofiber. In four-week-old wildtype mice, myofibers had a similar diameter without glycogen deposits (**F**). In PPD-stained section, no lipophilic inclusions were present (**G**). Ultrastructural images with regular sarcomere structures (**H**). At the subsarcolemmal level, few small vacuoles (**I**) and occasional empty vacuoles were present without glycogen accumulation (**J**). (Magnification: Resin sections × 40; TEM × 7000).

**Figure 3 cells-12-01602-f003:**
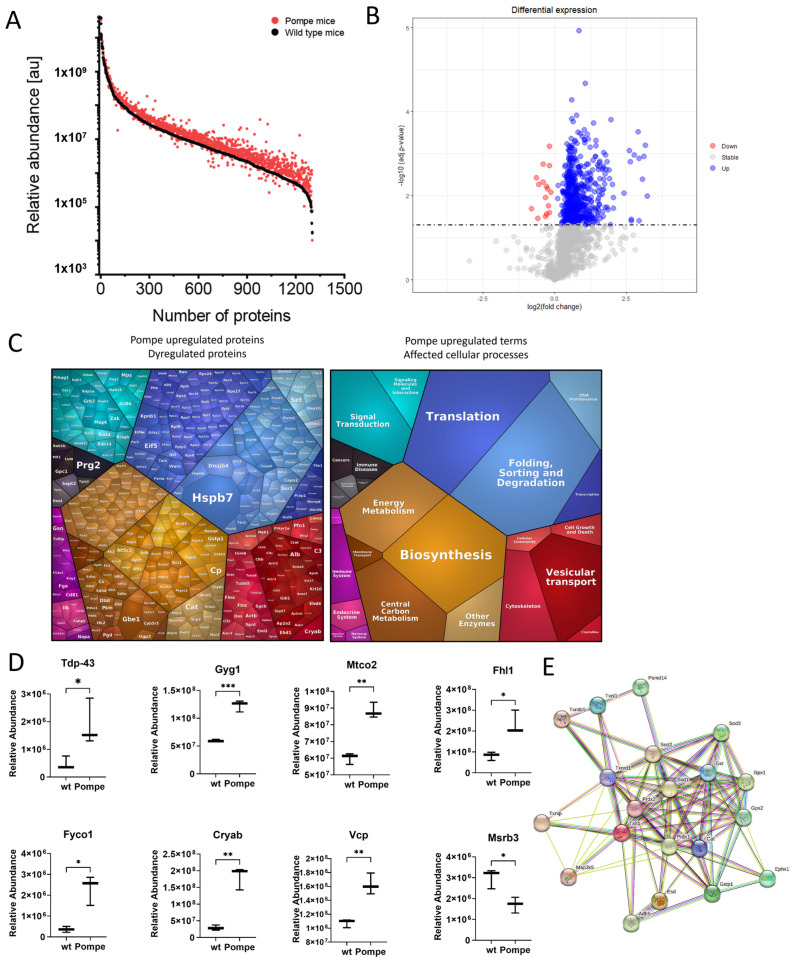
(**A**) Relative abundance of proteins (**B**) Volcano plot showed the abundance of all proteins that occurred in at least two wildtype and two Pompe mice. The threshold for significance was *p* < 0.05, with blue indicating significantly upregulated proteins and red indicating significantly downregulated proteins. (**C**) A Voronoi diagram with all significantly upregulated proteins. The left side shows protein names, with tile size being proportional to fold change. The right side shows the categorization of dysregulated proteins. (**D**) Boxplots showing a relative abundance of exemplary proteins involved in metabolism and proteostasis. With * *p* < 0.05, ** *p* < 0.01 and *** *p* < 0.001 (**E**) STRING analysis with significantly upregulated proteins involved in oxidative stress revealed Sod1 to be at the centre.

**Figure 4 cells-12-01602-f004:**
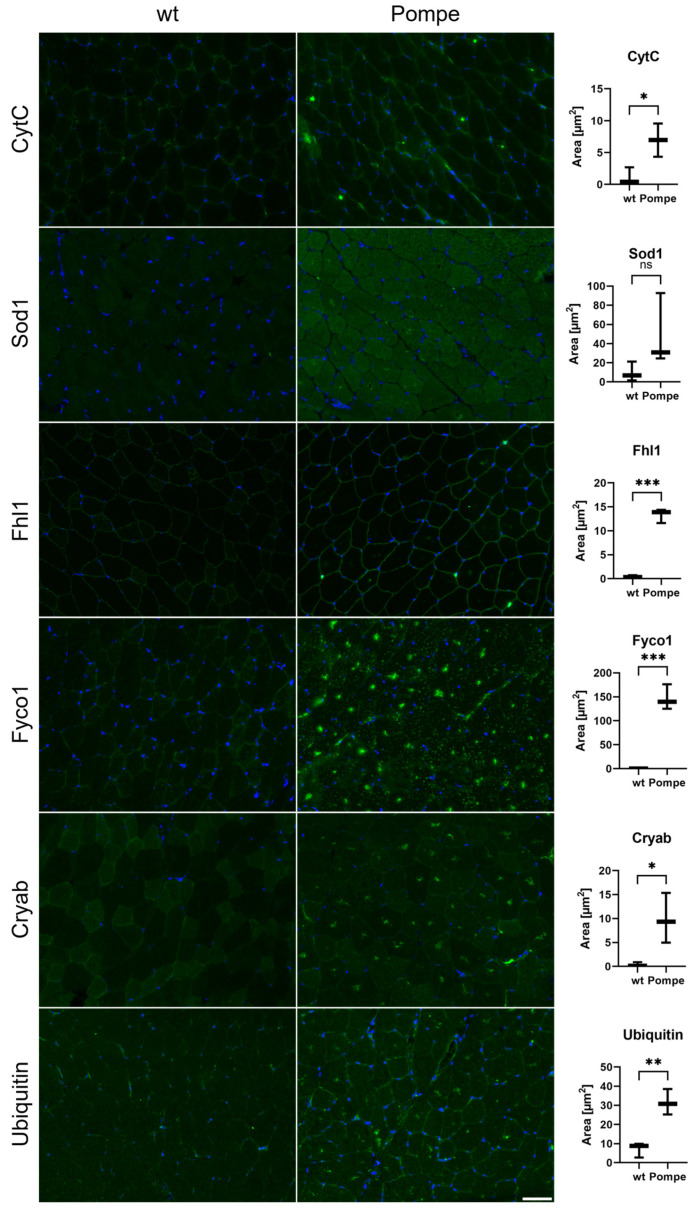
Immunofluorescence studies of quadriceps muscle sections in wildtype (wt) compared to Pompe mice. Sections were stained with antibodies against cytochrome c (CytC), Sod1, Fhl1, Fyco1, Cryab and ubiquitin (green). Nuclei were stained with DAPI (blue). Green-stained areas were quantified. Scalebar 50 µm. With * *p* < 0.05, ** *p* < 0.01 and *** *p* < 0.001.

**Figure 5 cells-12-01602-f005:**
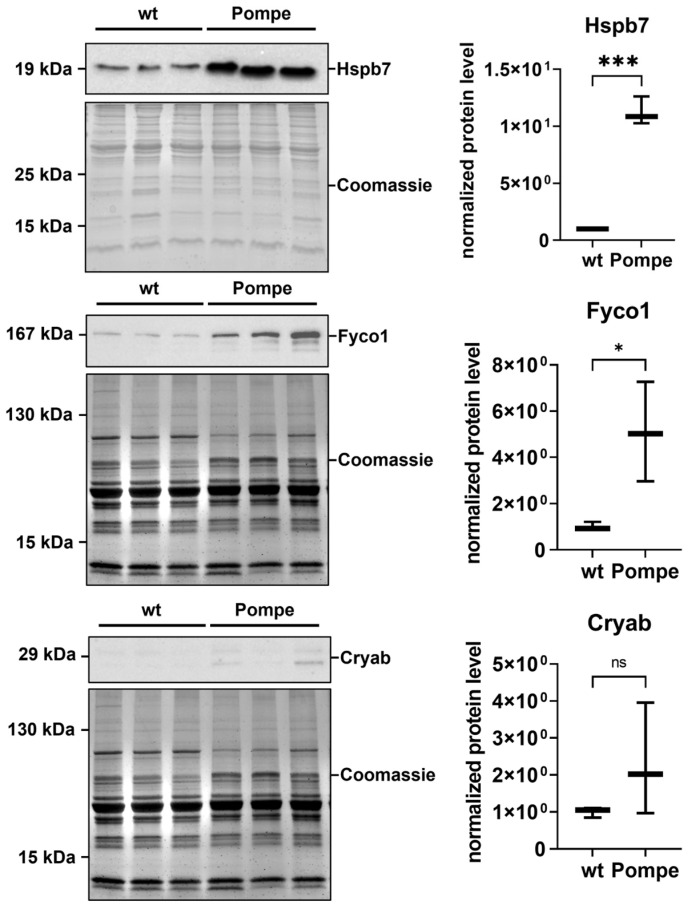
Western blot for validation of Hspb7, Fyco1 and Cryab in wildtype (wt) and Pompe mice. Left side shows the respective blots with total protein (Coomassie stained). Right side shows the quantification, normalized with total protein, on wildtype. With * *p* < 0.05 and *** *p* < 0.001.

**Table 1 cells-12-01602-t001:** Primary and secondary antibodies used for immunofluorescence study.

**Primary Antibody**	**Manufactory**	**Dilution**
anti-SOD1	GTX100554, Biozol, Eching, Germany	1:200
anti-CytC	sc-13156, Santa Cruz Biotechnology, Dallas, TX, USA	1:200
anti-FHL1	ab255828, Abcam, Cambridge, UK	1:300
anti-FYCO1	HPA035526, Sigma–Aldrich, St. Louis, MO, USA	1:500
anti-CRYAB	CPTC-CRYAB-2, deposited to the DSHB by Clinical Proteomics Technologies for Cancer	0.5 ng/µL
anti-Ubiquitin	ab19247, Abcam, Cambridge, UK	1:200
**Secondary Antibody**	**Manufactory**	**Dilution**
Alexa green 488 anti-rabbit	RRID: AB_2313584, Jackson ImmunoResearch Laboratories, Pennsylvania, PA, USA	1:1500
Alexa green 488 anti-mouse	RRID: AB_2307324, Jackson ImmunoResearch Laboratories, Pennsylvania, PA, USA	1:1000

## Data Availability

The data presented in this study are available on request from the corresponding author.

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
