# Peer review of "Dysregulation of Metabolism and Proteostasis in Skeletal Muscle of a Presymptomatic Pompe Mouse Model"

_cells, 2023, doi:10.3390/cells12121602_

Round 1

Reviewer 1 Report

The main of this study was to explore early proteomic changes in a mouse model of Pompe disease and identify potential therapeutic pathways using proteomic analysis of skeletal muscles from pre-symptomatic Pompe mice.

The research is very interesting in the field of understanding one of metabolic disorder pompe disease”.

Authors used the proteomic analysis of skeletal muscles for developing new diagnostic biomarkers for this disease. In addition, these biomarkers could help scientists in identify potential therapeutic pathways. This article contains most recent references, add some scientific sounds.

I invited authors to numbering the subtitles in material and methods.

I have some questions

What is the specific protein used COX1 or COX2?

Subtitles should be caplized in each word.

Title 3.5, is very long

Conclusion should be more clearly 

Please following the journal guidelines 

Please use the first letters of the author’s names in the author’s contribution section

Author Response

Point 1: I invited authors to numbering the subtitles in material and methods. Subtitles should be caplized in each word. Title 3.5, is very long. Please following the journal guidelines. Please use the first letters of the author’s names in the author’s contribution section.

Response 1: We thank the reviewer for these suggestions which were included in the revised version of the manuscript.

Point 2: What is the specific protein used COX1 or COX2?

Response 2: We wish to thank the reviewer for this valid question. In the proteomic dataset, Cytochrome c oxidase subunits Cox6c, COX4i1, COX6b1, COX2 (Mtco2) were found. Histological staining was done with CytC solution by Sigma (C2506). Immunofluorescence staining was carried out with the antibody cytochrome c (A-8) (sc-13156, Santa Cruz Biotechnology, Texas, USA), which is, according to manufacturer, raised against amino acids 1-104 of cytochrome c of equine origin.

Point 3: Conclusion should be more clearly.

Response 3: The conclusion section was rephrased accordingly.

Reviewer 2 Report

Everything about the paper is okay by me. However, since such research is considered preliminary or basic and is not validated by other researchers of similar expertise, the authors should change the statement of strongly recommending the inclusion of Pompe disease in the newborn screening program to avoid misleading the readers and scientific community at large.

The research is well-designed, and the authors have done justice to the topic. However, the conclusion should be reviewed and soften the recommendation made.

Author Response

Point 1: Everything about the paper is okay by me. However, since such research is considered preliminary or basic and is not validated by other researchers of similar expertise, the authors should change the statement of strongly recommending the inclusion of Pompe disease in the newborn screening program to avoid misleading the readers and scientific community at large. The research is well-designed, and the authors have done justice to the topic. However, the conclusion should be reviewed and soften the recommendation made.

Response 1: Thank you for your recommendation. As you are right about the study being on basic research, we now considered this important aspect when claiming that Pompe disease should be part of the newborn screening (see conclusion section).

Reviewer 3 Report

In this manuscript, Rohm and colleagues show the pathology of quadriceps muscle of Pompe mice at an early stage as well as the differential expression of proteins between wildtype and Pompe mice. The results provide novel insights in disease pathophysiology and potential guidance for future research. My major comments are as follows:

1.     Figure 2: control is missing. It should be provided concurrently with the Pompe mice.

2.     Figure 4: The images' resolution is poor. There should be a high-power image in the inset. Second, a Western blot analysis of these protein abundances should also be provided due to the absence of antibody specificity control.

Minor comments:

1.     What 16 proteins were shown to be downregulated by proteomics? There should be a detailed list of the proteins discovered by proteomics (both those that are upregulated and those that are downregulated).

2.     In Figures 1 and 4, the authors utilize immunofluorescence to measure the enzymes activity. When referring to enzyme activity, care should be taken because cofactors also play a role. A change in protein level may not always imply a change in activity. The writing needs to be modified.

Author Response

Point 1: Figure 2: control is missing. It should be provided concurrently with the Pompe mice.

Response 1: We thank the reviewer for this suggestion and will include a control image for electron microscopy.

Point 2: Figure 4: The images' resolution is poor. There should be a high-power image in the inset.

Second, a Western blot analysis of these protein abundances should also be provided due to the absence of antibody specificity control.

Response 2: The images with full resolution will be provided for the revised manuscript. Secondary antibody specifity was checked with Pompe and wildtype mice. Nevertheless, within the short time given for review we will provide additional Western Blots for proteins Fyco1 and Cryab for verification.

Point 3: What 16 proteins were shown to be downregulated by proteomics? There should be a detailed list of the proteins discovered by proteomics (both those that are upregulated and those that are downregulated).

Response 3: A list of all proteins significantly up- or downregulated will be included within the supplements of the manuscript.

Point 4: In Figures 1 and 4, the authors utilize immunofluorescence to measure the enzymes activity. When referring to enzyme activity, care should be taken because cofactors also play a role. A change in protein level may not always imply a change in activity. The writing needs to be modified.

Response 4: Thank you for raising that important point. We crosschecked the manuscript but did not find descriptions of immunofluorescence for measuring enzyme activity. We merely used the fluorescence to show the localization and abundance of proteins. The wording “Immunoreactivity” might be misleading and will be changed in the manuscript.

Round 2

Reviewer 3 Report

The issues are addressed.